# Mechanism of Resveratrol-Induced Programmed Cell Death and New Drug Discovery against Cancer: A Review

**DOI:** 10.3390/ijms232213689

**Published:** 2022-11-08

**Authors:** Jung Yoon Jang, Eunok Im, Nam Deuk Kim

**Affiliations:** Department of Pharmacy, College of Pharmacy, Research Institute for Drug Development, Pusan National University, Busan 46241, Korea

**Keywords:** resveratrol, apoptosis, autophagy, necroptosis, resveratrol derivatives and analogues, molecular mechanisms

## Abstract

Resveratrol (3,5,4′-trihydroxy-*trans*-stilbene), a polyphenol found in grapes, red wine, peanuts, and apples, has been reported to exhibit a wide range of biological and pharmacological properties. In addition, resveratrol has been reported to intervene in multiple stages of carcinogenesis. It has also been known to kill several human cancer cells through programmed cell death (PCD) mechanisms such as apoptosis, autophagy, and necroptosis. However, resveratrol has limitations in its use as an anticancer agent because it is susceptible to photoisomerization owing to its unstable double bond, short half-life, and is rapidly metabolized and eliminated. *Trans*-(*E*)-resveratrol is nontoxic, and has several biological and pharmacological activities. However, little is known about the pharmacological properties of the photoisomerized *cis*-(*Z*)-resveratrol. Therefore, many studies on resveratrol derivatives and analogues that can overcome the shortcomings of resveratrol and increase its anticancer activity are underway. This review comprehensively summarizes the literature related to resveratrol-induced PCD, such as apoptosis, autophagy, necroptosis, and the development status of synthetic resveratrol derivatives and analogues as novel anticancer drugs.

## 1. Introduction

Cancer is the second leading cause of death in the United States and an important public health problem worldwide [1]. According to the International Agency for Research on Cancer, the burden of cancer is increasing worldwide. Without effective intervention, 22 million new cancer cases are expected to occur annually by 2030, and the number of cancer deaths during the same period is expected to rise to 13 million [2]. Radiation, surgical removal of the tumor, and chemotherapy are some of the most common cancer treatment options; however, these traditional treatments have side effects [3]. Vomiting and nausea, chronic pain, loss of appetite, paresthesia, and fatigue are among the treatment-related side effects experienced by cancer patients [4,5]. In addition, current cancer treatments such as radiation therapy, chemotherapy, immune checkpoint inhibitors, and molecular targeted therapy are limited by the development of resistance to these treatments [6]. Therefore, it is necessary to develop new strategies or new treatments to overcome resistance. Researchers have mainly focused on medicinal plants or natural compounds with anticancer properties to overcome the complications and resistance to currently used drugs. Natural compounds derived from plants or other sources are of great interest in cancer prevention and treatment because of their safety and low toxicity [3].

Resveratrol (3,5,4′-trihydroxy-*trans*-stilbene) is a natural polyphenolic compound found in many plants, including foods such as grapes (particularly skin), peanuts, blueberries, and red wine. Resveratrol has a variety of biological effects, including anticancer, antioxidant, cardioprotective, anti-inflammatory, neuroprotective, and antidiabetic activities [7]. Resveratrol has been reported to have cytotoxic effects on numerous tumor cells [3].

In this review, we present the types of programmed cancer cell death induced by resveratrol and synthetic derivatives and analogues. These include type I cell death (apoptosis), type II cell death (autophagy dependence), and programmed necrosis (necroptosis).

## 2. Resveratrol

Resveratrol is a member of the stilbene family with two phenolic rings linked by an ethylene bridge. Resveratrol has two isomeric forms: *cis-(Z*)-resveratrol and *trans-(E*)-resveratrol [3]. The *trans* isoform is the main isoform, and is the most widely studied. However, upon exposure to ultraviolet, *cis* isomers are formed, which is referred to as photoisomerization [8] (Figure 1). Resveratrol is a phytoalexin produced by plants in response to attack or injury by pathogens such as bacteria and fungi [6]. Resveratrol was first isolated from the roots of the white hellebore (*Vinburnum grandiflorum*) in 1939 [9]. Resveratrol is mostly found in grapes, red wine, and peanuts [10]. The highest concentrations of resveratrol are found in *Polygonum japonicum* (formerly known as *Polygonum cruspdatum*), a Japanese knotweed used in oriental medicine and in various tea products [11]. The resveratrol concentration of red wine is notably higher than that of white wine, which is due to the difference in the wine fermentation process. The known concentrations of resveratrol in conventional foods are: red wine, 0.84–7.33 mg/L [12]; rosé wine, 0.29 mg/L [13]; white wine, 0–1.089 mg/L; red grapes, 92–1604 μg/kg fresh weight; white grapes, 59–1759 μg/kg fresh weight [14]; apples, 400 μg/kg fresh weight [15]; skin of tomatoes, ~19 μg/g dry weight [16]; peanuts without seed coats, 0.03–0.14 μg/g [17]; beers, 1.34–77.0 μg/L [18]; dark chocolate, 350 μg/kg; milk chocolate, 100 μg/kg [19]; and Itadori tea, 68 μg/100 mL [20].

## 3. Physiological Functions of Resveratrol

Resveratrol has been shown to have a variety of health-promoting effects, including anticancer [21,22,23], anti-aging [24,25,26], anti-inflammatory [25,27,28], neuroprotective [29,30,31], hepatoprotective [32,33,34], cardioprotective [35,36,37], antidiabetic [38,39,40], and antioxidant activity [41,42,43] (Figure 2). Resveratrol is a component of red wine, and is often thought to be an important factor in the French paradox, whereby the French population has a reduced risk of cardiovascular disease despite a high intake of saturated fat [44]. Resveratrol is also known to protect hepatic cells from oxidative stress by increasing antioxidant enzyme activity and altering gene expression [36]. In addition, it is reported to affect the expression and activity of diverse endogenous antioxidant enzymes [45], and to enhance urine protein excretion, renal dysfunction, and renal oxidative stress [46]. More significantly, resveratrol has been shown to sensitize many cancer cells resistant to anticancer drugs by overcoming the mechanism of chemical resistance [47], and thus promotes the efficacy of anticancer drugs at low doses [48].

## 4. Resveratrol in Cancer Therapy

Jang et al. [49] found, in 1997, that resveratrol prevented carcinogenesis in a mouse skin cancer model. Resveratrol exerts its effects at different cancer stages, from initiation and promotion to progression, by affecting various signaling pathways that regulate cell growth and division, apoptosis, inflammation, metastasis, and angiogenesis [6]. Furthermore, it has been shown that resveratrol protects normal cells while causing cancer cell death. This may be related to the targeting of different molecular and metabolic pathways in normal and cancer cells. Additionally, this double pattern of resveratrol action is dose-dependent [50]. Lower concentrations/doses (5–20 μM) increase the expression of cell survival proteins, while higher concentrations/doses (10–40 mM) stimulate apoptosis or necrosis [51,52]. High doses of resveratrol inhibit the synthesis of nucleic acids and proteins, cause damage to chromatin structures, and eventually lead to cell death [53].

According to Clinical-Trials.gov (accessed on 28 October 2022), a human clinical research database, several studies of resveratrol and cancer have been completed, and there are ongoing studies. According to the NCT00256334 study, grape powder containing low-dose resveratrol along with other bioactive components inhibited the expression of Wnt target genes, cyclin D1, and Axin in normal colon mucosa, suggesting that Wnt pathway inhibition could contribute to preventing resveratrol-mediated colon cancer. According to the NCT00433576 study, resveratrol was given 8 days before colectomy, and blocked M-1G adduct and cyclooxygenase-2 protein/Ki67 levels, potentially terminating tumor cell growth. The NCT00920803 study showed that resveratrol was detectable in hepatic tissue after administration of micronized resveratrol (SRT501). Cleaved caspase-3, a marker of apoptosis, was notably increased by 39% in malignant hepatic tissues after SRT501 treatment compared to tissues from placebo-treated patients. These clinical results indicate that resveratrol has potential for cancer treatment.

## 5. Effect of Resveratrol on Apoptosis

### 5.1. Apoptosis

Programmed cell death (PCD), also referred to as apoptosis or type I PCD, is one of the best characterized types of cell death. Apoptosis is closely linked to cellular processes that play a significant role in the development and homeostasis of multicellular organisms [54]. As tissue homeostasis entails a balance between apoptosis and cell proliferation, disruption of this balance (e.g., uncontrolled apoptosis) can be involved in a diverse range of human diseases, including cancer [55]. Apoptosis is characterized by chromatin condensation, cell shrinkage, chromosomal DNA fragmentation, cell membrane blebbing, and nuclear fragmentation [56]. It is executed primarily through intrinsic (mitochondrial) and extrinsic (death receptor) pathways.

### 5.2. Types of Apoptosis

#### 5.2.1. Intrinsic (Mitochondrial) Pathway

Intrinsic apoptotic pathways (mitochondria-dependent) involve intracellular signals that are activated in response to a variety of stress conditions (i.e., treatment with chemotherapeutic agents, irradiation, etc.) and converge at the mitochondria level [57]. Internal stimuli, for example, hypoxia, irreversible genetic damage, very high concentrations of cytosolic Ca^2+^, and excessive oxidative stress, are some of the factors activating intrinsic mitochondrial pathways [58]. Activation of proapoptotic BH3-only members of the Bcl-2 family (Bak, Bax) disrupts mitochondrial outer membrane permeability (MOMP) by neutralizing the antiapoptotic proteins Bcl-xL, Bcl-2, and Mcl-1. Thereby, proteins normally trapped in the intermembrane space are released into the cytoplasm. These proteins, also called apoptogenic factors, for example, cytochrome *c*, play an important role in activating mitochondria-dependent death in the cytosol [59]. Cytochrome *c* binds to the cytosolic apoptotic protease activating factor-1 (Apaf-1) and induces the formation of a complex called an apoptosome, which recruits the initiator procaspase-9 through the caspase recruitment domain (CARD), enabling proteolysis after autoactivation [60]. This process activates downstream executor caspase-3, -6, and -7 for cleavage of cellular proteins, resulting in apoptotic cell death [61].

#### 5.2.2. Extrinsic (Death Receptor) Pathway

The extrinsic apoptotic pathway (death-receptor-dependent) begins with the interaction of tumor necrosis factor (TNF) family ligands with cell surface death receptors that belong to the TNF receptor (TNFR) superfamily. Death receptors are structurally defined by an intracellular protein–protein interaction domain, called the death domain (DD), that is important for signal transduction that induces apoptosis [57]. The most characterized systems of death ligands and receptors include TRAILR1 (DR4)-TRAIL, TRAILR2 (DR5)-TRAIL, TNFR1-TNFα, and FAS (CD95, APO-1)-FasL. Upon stimulation of the death receptor by the corresponding ligand, the receptor and the cytoplasmic DD oligomerize, supporting homomorphic interactions with other DD-containing proteins. The role of adapter proteins (FADD/TRADD) is to recruit initiator procaspase-8 and/or -10 at this protein complex to induce the formation of the death-inducing signaling complex (DISC), increase the local concentration of procaspase, and promote mutual activation [62]. Activation of the initiator caspase results in the activation of downstream effector caspase-3, -6, and -7, which leads to the cleavage of substrates essential for cell viability, leading to cell death [63]. Some cells do not die via the extrinsic pathway alone, and need an amplification step induced by caspase-8. In this situation, caspase-8 cleaves BH3-only protein Bid (BH3-interacting-domain death agonist) to produce activated fragment t-Bid, which directly activates proapoptotic multidomain proteins to induce MOMP, thus connecting the extrinsic with the intrinsic pathways [58].

### 5.3. Induction of Apoptosis by Resveratrol

Based on the important role of apoptosis, its rational targeting may have a significant impact on cancer treatment, and if any agents, such as resveratrol, can regulate this pathway, it will provide a new means of cancer treatment. Various molecular targets of apoptosis regulated by resveratrol are summarized in Table 1.

#### 5.3.1. Effect of Resveratrol on Tumor Suppressor p53

p53 is a tumor suppressor that modulates cellular responses to genotoxic stress [22]. Activation of p53 induces the transcription of target genes involved in cell cycle arrest, DNA repair, and apoptosis [112]. p53 is a proapoptotic mediator that can activate the transcription of proapoptotic genes that encode Bcl-2 family members, such as phorbol-12-myristate-13-acetate-induced protein 1 (Noxa), Bcl-2-associated X protein (Bax), and p53-upregulated modulator of apoptosis (PUMA). However, p53 can also promote caspase activation by suppressing antiapoptotic genes, such as survivin, and upregulating the expression of apoptosis-inducing gene products, including Fas, Bid, DR5, and Apaf-1 [113]. p53 is primarily regulated through a variety of post-translational modifications, including phosphorylation, acetylation, methylation, neddylation, and ubiquitination [22].

Resveratrol can target the p53-mediated pathway to induce apoptosis. Liu et al. [22] showed that resveratrol inhibits cell viability in colorectal cancer cells, including HCT116, CO115, and SW480. They reported that resveratrol enhanced the expression of p53 and p53 target genes, including Bax and PUMA, which play a pivotal role in p53-dependent apoptosis. Furthermore, they demonstrated that treatment of cells with resveratrol upregulated the SET domain, including lysine methyltransferase 7/9 (SET7/9) expression, and positively regulated p53 through monomethylation at lysine 372. Collectively, these results demonstrate the molecular mechanism by which resveratrol induces p53 stability in colorectal cancer, thereby activating p53-mediated apoptosis. Applying bioinformatics analysis, Fan et al. [96] showed that resveratrol-treated A549 non-small-cell lung cancer cells were enriched in genes involved in apoptosis- or autophagy-related biological functions and p53 signaling pathways. In addition, resveratrol significantly reduced cell viability and increased the number of apoptotic cells, upregulating Bax expression and cleaved caspase-3 levels, while downregulating Bcl-2 expression levels. Furthermore, resveratrol increased p53 expression levels in a dose-dependent manner. It reduced p-MDM2 (at Ser166), an important negative regulator of p53 tumor suppressor, and decreased expression levels of its upstream regulator, p-Akt (at Ser473). Incubation of A549 cells with PFT-α, a specific inhibitor of p53, decreased the expression of the proapoptotic molecule Bax, and increased that of the antiapoptotic molecule Bcl-2, suggesting that the p53 pathway is involved in resveratrol-induced apoptosis and autophagy in these cells.

Mutations in p53 have been observed in more than 50% of human tumor tissues. Loss of p53 function in certain tumor types is associated with chemoresistance, and cancers with p53 mutations are generally poorly responsive to therapeutic agents, thus necessitating the search for anticancer agents that act independently of p53 status [114]. García-Zepeda et al. [78] showed that resveratrol treatment increased cell cycle arrest in the G1 phase in cervical cancer cells carrying the p53 mutation C33A, i.e., HeLa (HPV18-positive), CaSki, and SiHa cell lines (HPV16-positive). It also induced apoptosis in all cell lines, especially in CaSki cells. In addition, the mitochondrial membrane potential that appears when apoptosis is induced in HeLa, CaSki, and SiHa cells was reduced by resveratrol treatment. Resveratrol increased the expression of the p53 protein carrying the p53 mutation C33A in CaSki and SiHa cell lines (both HPV16-positive). In contrast, resveratrol treatment decreased p53 expression in CaLo and HeLa cells (both HPV18-positive). These results suggest that HPV18-positive cell lines may die by a mechanism independent of p53. Gogada et al. [70] showed that treatment of MDA-MB-231 breast cancer cells with resveratrol induced a p53-independent X-linked inhibitor of apoptosis protein (XIAP)-mediated translocation of Bax into mitochondria that underwent oligomerization to initiate apoptosis. Resveratrol treatment facilitated the interaction between the Bax and XIAP in cytosol and mitochondria, suggesting that XIAP plays an important role in the activation of Bax and its potential transfer to the mitochondria. This process did not include p53, but needed the accumulation of Bim and t-Bid in mitochondria. Bax underwent homo-oligomerization mainly in the mitochondria, and played an important role in the release of cytochrome *c* into the cytosol. Another key protein regulating mitochondrial membrane permeability, Bak, did not interact with p53, but continued to bind with Bcl-xL. Taken together, these authors demonstrated that resveratrol induces Bax-dependent, but p53-independent, apoptosis in breast cancer cells.

#### 5.3.2. Effect of Resveratrol on AKT

AKT serine/threonine kinase, also recognized as protein kinase B (PKB), is an oncogenic protein that controls apoptosis, cell survival, growth, proliferation, and glycogen metabolism [115]. AKT plays an important role in the phosphoinositide 3-kinase (PI3K)/AKT signaling pathway. It is activated by inflammation, growth factors, and DNA damage, as well as PI3K or phosphoinositide-dependent kinase (PDK). Signal transduction occurs via downstream effectors such as mammalian target of rapamycin (mTOR), forkhead box protein O1 (FOXO1) or glycogen synthase kinase 3 beta (GSK3β). Abnormal overexpression or activation of AKT has been reported in many cancers, including lung, ovarian, and pancreatic cancers, which are associated with increased cancer cell proliferation and survival. Consequently, targeting AKT may provide a significant approach for cancer prevention and treatment [116].

Wang et al. [109] showed that resveratrol inhibited PC-3 prostate cancer cell growth and induced apoptosis. In addition, resveratrol affected the expression of epithelial–mesenchymal-transition-related proteins through an increase in E-cadherin and decreased vimentin expression. In addition, resveratrol inhibited Akt phosphorylation in PC-3 cells. After the addition of an Akt inhibitor, apoptosis was notably promoted compared to the resveratrol-treated group. However, after the addition of an Akt activator, apoptosis was remarkably inhibited compared to the resveratrol-treated group, and changes similar to the control group were observed. When an Akt inhibitor was added, the expression of Bax, caspase-3, and caspase-9 mRNA was increased, but the expression of Bcl-2 mRNA was decreased compared to the resveratrol treatment group. In addition, the Akt inhibitor upregulated the mRNA levels of E-cadherin, an epithelial marker, while the expression of vimentin, a mesenchymal marker, was remarkably decreased. However, the Akt activator outstandingly reversed all these resveratrol-induced effects. Taken together, these findings show that resveratrol promotes apoptosis and inhibits epithelial–mesenchymal transition in PC-3 cells by inhibiting the PI3K/Akt signaling pathway.

Takashina et al. [86] showed significant differences in the anti-viability effect of resveratrol between different types of human cancer cells. Resveratrol considerably inhibited the viability of U937 and MOLT-4 leukemia cells, had a moderate inhibitory effect on the viability of MCF-7 breast, HepG2 liver and A549 lung cancer cells and a slight impact on the survival of Caco-2, HCT116 and SW480 colorectal cancer cells. After resveratrol treatment, the population of late apoptotic U937 and MOLT-4 cells was significantly increased, whereas the population of early apoptotic MCF-7 and HepG2 was increased, and resveratrol-induced DNA fragmentation was only observed in leukemia cells. In addition, resveratrol notably reduced Akt activation by downregulation of H-Ras, promoting Bax translocation to the mitochondria in leukemic cells.

Li et al. [71] showed that resveratrol-mediated inhibition of cell proliferation was associated with apoptosis induction and G1 phase cell cycle arrest in DLD1 and HCT15 colorectal cancer cells. Resveratrol treatment reduced protein expression levels of cyclin D1, cyclin E2 and Bcl-2 apoptosis regulators, and increased the levels of Bax and p53, all of which are involved in the regulation of the cell cycle and apoptosis. In particular, the results obtained from in silico computer screening identified AKT1 and AKT2 as novel targets of resveratrol. Computer docking suggested that the active pockets of AKT1 and AKT2 have three or four possible hydrogen bonds contributing to the mode of action of resveratrol. They also confirmed, using pull-down analysis, that resveratrol binds to AKT1 and AKT2. In addition, knockdown of AKT1 and AKT2 inhibited cell proliferation and colony growth by attenuating cell cycle progression and upregulating the apoptosis of colorectal cancer cells, similar to resveratrol treatment. Thus, these authors demonstrated that the targeting of AKT1 and AKT2 by resveratrol could be a powerful strategy for chemoprevention or treatment for colorectal cancer.

#### 5.3.3. Effect of Resveratrol on SIRT

Sirtuin belongs to class III histone deacetylases (HDACs) and is a NAD^+^-dependent enzyme. These enzymes are involved in significant cellular processes, including cell survival, metabolism, stress response, senescence, aging, and tumorigenesis via deacetylation of several substrates [117]. There are seven sirtuin isoforms in mammals designated as SIRT1-7. They participate in a broad range of cellular processes and pathways with diverse cellular localizations and molecular targets. SIRT1 normally protects cells from oncogenic transformation. However, SIRT1 enzymatic activity can also promote cancer growth by the inactivation of proapoptotic factors. SIRT1 has a dual role; it can promote or inhibit cancer depending on the cellular context, specific signaling pathways, or specific cancer targets [118].

Chao et al. [66] showed that resveratrol notably reduced cell viability and induced apoptosis in JJ012 chondrosarcoma cells in a dose-dependent manner. Protein expression and activity of SIRT1 were increased after resveratrol treatment. Resveratrol remarkably inhibited NF-κB signaling by deacetylating the p65 subunit of the NF-κB complex, which was reversed by siRNA-SIRT1 transfection. Resveratrol-caused apoptosis involves a caspase-3-mediated mechanism. Both siRNA-SIRT1 transfection and the deacetylation inhibitor MS-275 prevented resveratrol-induced caspase-3 cleavage and activity in JJ012 chondrosarcoma cells. Moreover, an in vivo xenograft study showed resveratrol treatment resulted in a dramatic reduction in tumor volume and increased SIRT1 and cleaved caspase-3 levels. These results demonstrate that resveratrol causes chondrosarcoma cell apoptosis through SIRT1-activated NF-κB deacetylation and shows anti-chondrosarcoma activity in vivo.

Jin et al. [67] reported that resveratrol reduced cell viability and induced apoptosis of SW1353 chondrosarcoma cells in a dose-dependent manner. The levels of cleaved caspase-3, SIRT1, and Bax were upregulated, and the expression levels of Bcl-2 and phosphorylation of signal transduction and activator of transcription 3 (STAT3) were downregulated. Furthermore, they showed that resveratrol treatment activated SIRT1, but impaired resveratrol’s ability to suppress STAT3 phosphorylation in cells transfected with SIRT1-siRNA. They demonstrated that resveratrol treatment induces apoptosis, prevents proliferation, and affects phosphorylation of STAT3 by activating SIRT1 in SW1353 chondrosarcoma cells.

## 6. Effect of Resveratrol on Autophagy

### 6.1. Autophagy

Derived from the Greek “self-eating”, the term autophagy describes the catabolic process by which cytoplasmic cargo is delivered to lysosomes for degradation and recycling, and is a type II PCD [119]. Autophagy and the ubiquitin–proteasome system (UPS) are two important cellular proteolytic pathways essential for maintaining cellular protein homeostasis. However, while the UPS is primarily dedicated to the rapid degradation of short-lived proteins, autophagy degrades long-lived proteins and organelles [120]. Under basal conditions, autophagy contributes to the metabolic balance of cells, ensuring cell viability. After lysosomal degradation, degradation products are released into the cytoplasm and recycled to generate energy [121]. However, under stressful conditions, such as hypoxia, nutrient deprivation, genotoxic stress, and pathogen infection, autophagy maintains cellular homeostasis, removes misfolded or aggregated proteins, removes damaged organelles, such as mitochondria and endoplasmic reticula, and clears intracellular pathogens. By eliminating these components, it is dramatically induced to ensure proper quality control [122]. However, insufficient or excessive autophagy can lead to cell death. Thus, autophagy defects are implicated in the pathogenesis of diverse diseases, such as cancer and neurodegenerative disorders [123].

In cancer biology, autophagy is considered a double-edged sword; it can act as a tumor suppressor and a tumor promoter, depending on the stage of carcinogenesis. In early tumorigenesis, autophagy inhibits tumor initiation and suppresses cancer progression through survival pathways and quality control mechanisms. However, as tumors progress and reach terminal stages, autophagy serves as a dynamic degradation and recycling system that promotes the survival and growth of established tumors and, ultimately, metastasis. This shows that modulation of autophagy can be used as an efficient interventional strategy for cancer therapy [124].

### 6.2. Types of Autophagy

There are three types of autophagy according to the method of delivery to the lysosome: macroautophagy, microautophagy, and chaperone-mediated autophagy (CMA) [125]. Macroautophagy is the best characterized form of autophagy, and is commonly referred to simply as autophagy. Macroautophagy is a process by which cells form double-membrane autophagic vesicles, called autophagosomes, that sequester misfolded proteins and damaged organelles and fuse with lysosomes to form autolysosomes for degradation [126]. Microautophagy refers to the process by which damaged organelles or misfolded proteins are directly packaged by lysosomal traps without forming autophagosomes [127]. CMA stands for chaperone-dependent selection of soluble cytoplasmic proteins to be targeted to lysosomes. They translocate across the lysosomal membrane for degradation. A unique characteristic of this type of autophagy is the selectivity for the degraded proteins and the direct movement of these proteins without requiring the formation of additional vesicles [119]. Some recent studies have emphasized the important roles of microautophagy and CMA in tumor growth and progression. However, almost all studies of the role of autophagy in cancer progression, development, and treatment refer to macroautophagy [54].

### 6.3. Induction of Autophagy by Resveratrol

The final goal of all antitumor therapies is to kill tumor cells effectively and specifically. Until now, the main method of antitumor therapy has been to induce apoptosis. However, tumor cells can escape apoptosis through several pathways triggered by antitumor therapy, among which autophagy is of increasing interest [127]. The various molecular targets of autophagy regulated by resveratrol are summarized in Table 2.

#### 6.3.1. Effect of Resveratrol on AMPK

5′ Adenosine monophosphate-activated protein kinase (AMPK) is an evolutionarily conserved serine/threonine protein kinase, which functions as an energy sensor and plays an essential role in catabolism upregulation and anabolism inactivation. Under diverse physiological and pathological conditions, AMPK can be phosphorylated by upstream kinases and activated by binding to adenosine monophosphate (AMP) or adenosine diphosphate (ADP) rather than adenosine triphosphate (ATP). Activated AMPK controls various metabolic processes, including autophagy. AMPK advances autophagy directly by phosphorylating autophagy-related proteins, including Unc-51-like autophagy activating kinase (ULK1), mTORC1, and phosphatidylinositol 3-kinase catalytic subunit type 3 (PIK3C3)/VPS34 complexes, or indirectly by controlling the expression of autophagy-related genes downstream of transcription factors such as transcription factor EB (TFEB), forkhead box O3 (FOXO3), and bromodomain-containing protein 4 (BRD4) [137]. AMPK may act as a tumor growth suppressor or promoter, depending on the cancer type and circumstances [8].

Ma et al. [99] reported that resveratrol promotes the cleavage of caspase-3 and PARP, which are apoptosis-related proteins, and reduces, dose-dependently, the protein levels of survivin in U266, RPMI-8226, and NCI-H929 multiple myeloma cells. Furthermore, resveratrol increases the levels of Beclin-1 and LC3 in a dose-dependent manner, indicating that autophagy may be involved in the anti-multiple myeloma effect of resveratrol. The autophagy inhibitor 3-methyladenine (3-MA), alleviated the cytotoxicity of resveratrol and decreased the levels of autophagy-related proteins LC3 and Beclin-1. Furthermore, 3-MA inhibited resveratrol-induced apoptosis and decreased PARP and caspase-3 cleavage associated with apoptosis. Resveratrol upregulated the phosphorylation of AMPKα and downregulated the phosphorylation of mTOR and its downstream substrates p70S6K and 4EBP1, resulting in autophagy.

Chang et al. [100] demonstrated that resveratrol induced autophagy and apoptosis in cisplatin-resistant CAR oral cancer cells. Resveratrol treatment increased the protein levels of important autophagy markers, including autophagy-related 5 (Atg5), Atg7, Atg12, Atg14, Atg16L1, Beclin-1, PI3K class III, and LC3-II, while it decreased Rubicon protein levels, indicating autophagy induction. In addition, resveratrol upregulated the protein levels of AMPKα phosphorylated at Thr172 and AMPKα, but downregulated phosphorylation of Akt at Ser473 and mTOR at Ser2448. These results show that resveratrol induced autophagic cell death via the modulation of AMPK and Akt signaling. Autophagy inhibitors (3-MA) and AMPK inhibitors (compound c) reduced LC3-II protein levels and improved cell viability. In addition, resveratrol increased protein levels of cleaved caspase-3 and -9, apoptosis-related protein molecules, such as cytochrome *c*, apoptotic protease activating factor-1 (Apaf-1), apoptosis-inducing factor (AIF), endonuclease G (Endo G), and Bax, but it decreased protein levels of Bcl-2 and the phosphorylation of Bcl-2-associated death promoter (Bad) at Ser136. The pan-caspase inhibitor Z-VAD-FMK weakened resveratrol-induced caspase-3 and -9 cleavage and apoptosis.

#### 6.3.2. Effect of Resveratrol on p62

p62 or sequestosome 1 (SQSTM1) is considered an effector of selective autophagy and an autophagy substrate [138]. p62 is involved in the degradation of cytotoxic substances through the autophagy–lysosomal system. The p62 protein interacts, via the LC3-interacting (LIR) domain, with LC3, and thus attaches to the autophagosome, where it delivers ubiquitinated cytotoxic substances, attached to it via the ubiquitin-associated domain (UBA), to the lysosome for degradation. Under physiological conditions, p62 functions as a signal transduction adapter, and its levels are relatively low because of continued degradation by autophagy [139]. p62 is a well-known and important regulator of selective autophagy that functions in the formation of cargoes in autophagic machinery [140].

Zhang et al. [133] showed that resveratrol induces autophagy and apoptosis in A549 non-small-cell lung adenocarcinoma cells. They demonstrated that resveratrol treatment increased autophagy and autophagy-mediated degradation of p62. They also indicated that p62 co-localized with the Fas/caveolin-1 (Cav-1) complex, which is known to be involved in apoptosis. However, siRNA-mediated p62 downregulation improved the formation of the Fas/Cav-1 complex, suggesting that p62 prevents its formation. Furthermore, they showed that the Fas/Cav-1 complex triggered caspase-8 activation and cleavage of Beclin-1, releasing the C-terminal Beclin-1 peptide that migrates to the mitochondria and initiates apoptosis. In addition, it was shown that p62 knockdown by siRNA increased the activation of caspase-8 and initiated apoptosis, whereas Cav-1 knockdown prevented apoptosis but increased autophagy. They suggested that p62 links resveratrol-induced autophagy to apoptosis. It was also concluded that p62 inhibits the formation of the Fas/Cav-1 complex to block apoptosis; however, RSV-induced autophagic degradation activated caspase-8-mediated Beclin-1 cleavage to initiate apoptosis by transferring the Beclin-1 C-terminal fragment to the mitochondria.

In another study, Puissant et al. [129] reported that resveratrol induces autophagy in imatinib-sensitive (IM-S) and imatinib-resistant (IM-R) K562 chronic myeloid leukemia cells through JNK-dependent accumulation of p62. They also showed that JNK inhibition or p62 knockdown inhibited resveratrol-mediated autophagy and antileukemia effects. Furthermore, resveratrol inhibited the mTOR pathway by stimulating AMPK. AMPK knockdown or mTOR overexpression inhibited resveratrol-induced autophagy but not JNK activation. Additionally, p62 expression and autophagy were induced by resveratrol in CD34+ precursor cells in chronic myelogenous leukemia patients, and inhibition of autophagy protected CD34+ chronic myelogenous leukemia cells from resveratrol-mediated cell death. In conclusion, these authors demonstrated that resveratrol induces autophagic cell death in chronic myelogenous leukemia cells through JNK-mediated p62 overexpression and AMPK activation.

## 7. Induction of Necroptosis by Resveratrol

Necroptosis, also known as programmed necrosis, is characterized by the activation of receptor-interacting serine/threonine protein kinases (RIPKs) through multiple signaling pathways [141]. Unlike apoptosis, necroptosis is unrelated to caspase [142]. RIPKs are activated when recruited into macromolecular complexes by various cell surface receptors, such as Toll-like receptors (TLRs), T-cell receptors (TCRs), and death receptor (DRs) [143]. The formation of necrosome, a complex composed of RIPK1, RIPK3, and mixed lineage kinase domain-like (MLKL) is central to the necroptosis mechanism [144]. RIPK3 further activates the downstream molecule MLKL through phosphorylation, leading to MLKL oligomerization [145,146]. Oligomerized MLKL is inserted into and penetrates the cell membrane, which eventually leads to cell death [147]. Necroptosis has pro-cancer and anticancer effects [55,148]. This dual effect of promotion and reduction of tumor growth has been observed in various cancer types [149,150,151]. Apoptosis of cancer cells was induced to eliminate malignant cells [152]. However, deregulation of apoptosis signaling in cancer, particularly the activation of antiapoptosis systems, allows cancer cells to avoid this program and leads to uncontrolled cell proliferation and tumor survival. Thus, tumor necroptosis, a form of caspase-independent cell death, has great therapeutic potential for cancer treatment [153].

Currently, there are not many studies on necroptosis and resveratrol. According to a study by Lee et al. [154], resveratrol effectively reduced the viability of LNCaP prostate carcinoma cells in a concentration- and time-dependent manner. They also showed that the combination of resveratrol and docetaxel induced synergistic cytotoxicity. Resveratrol alone, or in combination with docetaxel, induced G2/M phase arrest and DNA damage responses. Resveratrol increased the apoptosis-related Bax/Bcl-2 ratio, as well as caspase-3 and PARP cleavage, and these were further elevated with resveratrol and docetaxel. Additionally, resveratrol increased the levels of p-RIPK3 and p-MLKL as mediators of necroptosis. This resveratrol effect on necroptosis was synergistic to that of docetaxel. Suppressing necroptosis or apoptosis by pretreating with necrostatin-1 (necroptosis inhibitor) or Q-VD-Oph-1 (pan-caspase inhibitor), respectively, improved cell viability, and reversed the levels of apoptosis- or necroptosis-inducing molecules in LNCaP cells treated with resveratrol and docetaxel. This strongly suggests that cell death induced by resveratrol and docetaxel is mediated by the simultaneous induction of both apoptosis and necroptosis.

## 8. Cell Death Mechanism of Synthetic Resveratrol Derivatives and Analogues in Cancer

Resveratrol is very photosensitive and poorly soluble in water [155]. Although the anticancer effects of resveratrol have been shown, its clinical application is limited owing to its short biological half-life and rapid metabolism and clearance [156]. In humans and rodents, resveratrol is metabolized via three major pathways [157], and even high doses of resveratrol may be insufficient to achieve the systemic concentrations required for cancer prevention. After intestinal absorption, *trans*-(*E*)-resveratrol and its glucoside are converted to glucuronide and sulfate metabolites by enterocytes and hepatocytes via the action of uridine 5′-diphospho-glucuronosyltransferases and sulfotransferases, respectively [158]. Rapid sulfate conjugation in the intestine and liver appears to limit bioavailability of resveratrol [157]. Additionally, the gut microbiota has the potential to catalyze the hydrogenation of resveratrol aliphatic double bonds [159]. Therefore, many researchers show great interest in developing new formulations with improved resveratrol bioavailability along with potent anticancer activity. The molecular targets of synthetic resveratrol derivatives and analogues for the induction of PCD are summarized in Table 3.

### 8.1. Programmed Cell Death Induced by Synthetic Resveratrol Derivatives

Several studies have reported that synthetic resveratrol derivatives induce PCD in cancer cells (Figure 3). In connection with the development of resveratrol derivatives as anticancer agents, methoxylated stilbenes are emerging as a new class of apoptosis-inducing agents. Substitution of hydroxyl groups with methoxyl groups may increase the structural stability of resveratrol. The methoxyl resveratrol compounds are highly lipophilic, and therefore have better bioavailability than the parent compound. The glucuronidation and sulfidation of methoxyl derivatives are lower than that of the parent resveratrol in the metabolic process, confirming that methoxylation induces stronger bioactive responses [181]. Feng et al. [161] showed that, compared to the parent compound, the novel derivative 2,3′,4,4′,5′-pentamethoxy-*trans*-stilbene (PMS) had a stronger in vitro antimitogenic effect on HT29 colorectal cancer cells (Figure 3B). In addition, PMS prevented tumor growth in vivo in colon cancer xenograft models. In this regard, PMS strongly caused apoptosis in HT29 cells, as evidenced by increased DNA fragmentation, PARP cleavage, and the accumulation of cells in the sub-G1 phase of the cell cycle. These authors also showed that PMS enhanced microtubule polymerization and G2/M mitotic arrest and caspase-dependent apoptosis. Collectively, it was demonstrated that PMS is a potent inducer of apoptosis through microtubule targeting.

According to Park et al. [162], 3,5-diethoxy-3′,4′-dihydroxy-*trans*-stilbene (Res-006), another novel resveratrol derivative, inhibited the viability of HepG2 hepatoma cells more effectively than resveratrol (Figure 3D). Combination treatment with the endoplasmic reticulum stress (ER stress) modulator 4-phenylbutyrate or the ROS inhibitor N-acetyl-L-cysteine significantly attenuated Res-006-induced HepG2 cell death, which resulted in proapoptotic ER stress and/or ROS accumulation that was directly correlated with Res-006-induced HepG2 apoptotic cell death. They also found that treatment of HepG2 cells with Res-006 immediately induced dysregulation of mitochondrial dynamics and accumulation of mitochondrial ROS. It also disrupted mitochondrial membrane potential and induced ER stress and cell death. Collectively, these authors demonstrated that Res-006 can kill HepG2 cells via cell death pathways, including ER stress initiated by mitochondrial ROS accumulation.

### 8.2. Programmed Cell Death Induced by Synthetic Resveratrol Analogues

Numerous studies have reported that synthetic resveratrol analogues induce PCD in cancer cells (Figure 4). Unlike resveratrol, the synthetic resveratrol analogue 4-(6-hydroxy-2-naphtyl)-1,3-benzenediol) (HS-1793) does not have unstable double bonds, and instead has an aromatic ring with two different hydroxyl positions (Figure 4G). These structural changes are more metabolically stable, less photosensitive, and more potent than resveratrol [155]. Several studies have shown that HS-1793 is a well-known substance that induces apoptosis in several types of cancer. Kim et al. [171] showed that HS-1793 can inhibit cell proliferation and induce apoptosis in MCF-7 (wildtype p53) and MDA-MB-231 (mutant p53) cells with different p53 statuses. HS-1793 induced G2/M arrest and apoptosis by decreasing cyclin and CDK while increasing Bax, p53, and p21 in both cell lines. The effect was mediated through a p53-dependent or -independent pathway. In addition, HS-1793 showed a stronger cytotoxic effect than resveratrol on MCF-7 and MDA-MB-231 breast cancer cells.

Another study by Kim et al. [172] reported that HS-1793 prevented cell growth and induced apoptosis in HCT116 colorectal cancer cells in a concentration-dependent manner. Induction of apoptosis was shown by morphological changes, as well as changes in the Bax/Bcl-2 expression ratio, PARP cleavage, and caspase activation. It was also shown that HS-1793 induced G2/M arrest, and had a more potent anticancer effect than resveratrol. Furthermore, it was found that HS-1793 inhibited Akt, and the PI3K/Akt inhibitor LY294002 enhanced the effect of HS-1793 on apoptosis induction.

According to Um et al. [173], resveratrol treatment of HT29 colorectal cancer cells induced CCAAT/enhancer-binding protein (C/EBP) homologous protein (CHOP), ER-stress-specific XBP1 splicing, and glucose-related protein (GRP)-78. However, no such induction was observed in HS-1793-treated HT29. They also reported that, unlike resveratrol, HS-1793 did not induce ER-stress-mediated apoptosis, but decreased phosphorylated Akt levels. The robust antitumor activity of HS-1793 derives at least in part from its ability to inactivate Akt.

According to the study by Jeong et al. [174], in U937 leukemic cells, HS-1739 showed a stronger antitumor effect than resveratrol in most cancer cells. The resistance conferred by Bcl-2 represents an attractive therapeutic strategy for cancer, demonstrating that HS-1793 overcomes the resistance conferred by Bcl-2. HS-1793 treatment resulted in a significant downregulation of 14-3-3 proteins, suggesting that it overcomes the resistance conferred by Bcl-2 in U937 cells through 14-3-3. They also observed that HS-1793 exerts antitumor activity through Bad. Therefore, based on the results of several researchers, HS-1793 is potentially a candidate chemotherapeutic agent for several types of cancer.

Analysis of the structure–activity relationship showed that the methoxy groups at positions 3,5- and 3,4,5- of the stilbene backbone may be important for the proapoptotic effect of the compounds [182]. According to a study by Piotrowska et al. [176], the resveratrol analogue 3,4,4′,5-tetramethoxystilbene (DMU-212) showed stronger cytotoxic activity in DLD-1 colorectal cancer cells than in LOVO colorectal cancer cells (Figure 4I). Analysis of the expression patterns of 84 apoptosis-related genes showed that the two colorectal cancer cells expressed transcripts specific for the mitochondria-mediated apoptosis pathway. They found that DMU-212 upregulated the levels of proapoptotic Bak1, Bok, Bik, Noxa, Bad, Bax, p53, and Apaf1 transcripts in DLD-1 cell lines, while downregulated the levels of antiapoptotic Bcl-2, Bcl-xL, and Bag1 mRNA. In addition, changes in apoptosis-related gene expression were less pronounced in LOVO cells that did not express CYP1B1 protein, and had lower levels of CYP1A1 protein than in DLD-1 cells.

Another study by Piotrowska et al. [177] reported that DMU-212 arrested A-2780 and SKOV-3 ovarian cancer cells at the G2/M or G0/G1 phase of the cell cycle, resulting in apoptosis. In SKOV-3 cells, DMU-212 induced the upregulation of proapoptotic Bax, Apaf-1, and p53 genes specific for the intrinsic pathway of apoptosis, and decreased Bcl-2 and Bcl-xL mRNA expression. Conversely, it increased the expression of the proapoptotic genes Fas, FasL, TNF, TNFRSF10A, TNFRSF21, and TNFRSF16 specific for the extrinsic pathway of apoptosis in A-2780 cells. DMU-212 reduced CYP1A1 and CYP1B1 mRNA levels by 50% and 75% in A-2780 and by 15% and 45% in SKOV-3 cells, respectively, as well as the expression of the corresponding proteins in both cell lines. They reported that, unlike CYP1B1, expression patterns in the two ovarian cell lines could influence the susceptibility of DMU-212 to the cytotoxic activity.

According to a study by Jozkowiak et al. [178], DMU-212 undergoes metabolic oxidation, hydroxylation, and *O*-demethylation, resulting in four metabolites of which 4′-hydroxy-3,4,5-trimetoxystilbene (DMU-281) induced apoptosis in DLD-1 and LOVO colorectal cancer cells (Figure 4J). They showed that the cytotoxic activity of DMU-281 was triggered through cell cycle arrest at the G2/M phase and the induction of apoptosis through the activation of caspase-3, -7, -8, and -9. In addition, it was shown that DMU-281 altered the expression pattern of genes and proteins involved in the intrinsic and extrinsic pathways of apoptosis.

## 9. Conclusions

This review article investigates the mechanisms of PCD, namely apoptosis, autophagy, and necroptosis, following treatment with natural resveratrol and synthetic resveratrol derivatives and analogues as tumor suppressors in cancer (Figure 5). Resveratrol induces PCD in several human cancer cell lines in vitro, and has been reported to have clear anticancer effects. However, resveratrol is very photosensitive, poorly soluble in water, has a short biological half-life, and is rapidly metabolized and eliminated, thus limiting its clinical application. Therefore, several researchers have synthesized several synthetic resveratrol derivatives and analogues to ameliorate these shortcomings. These induced PCD, apoptosis, and autophagy in several cancer cells, exhibiting increased potency and/or a variety of selective activities compared to the parent compound resveratrol. Therefore, these novel synthetic resveratrol derivatives and analogues may be promising chemicals that specifically target a variety of cancer cells and induce PCD. However, synthetic resveratrol derivatives and analogues require additional molecular mechanism studies, animal experiments, and preclinical studies. Moreover, additional information is needed, including data on bioavailability and safety profiles in clinical studies. These additional studies suggest that synthetic resveratrol derivatives and analogues offer great potential as cancer therapeutics. Accordingly, resveratrol could be a significant lead compound for the development of new anticancer drugs.

## Figures and Tables

**Figure 1 ijms-23-13689-f001:**
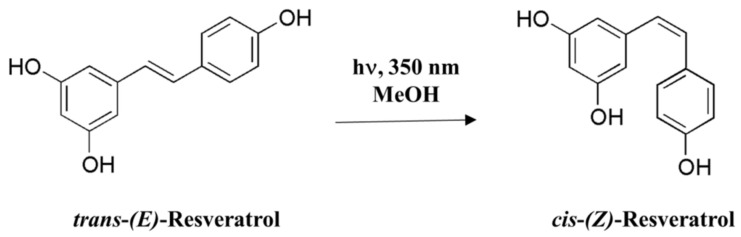
Resveratrol isomerization.

**Figure 2 ijms-23-13689-f002:**
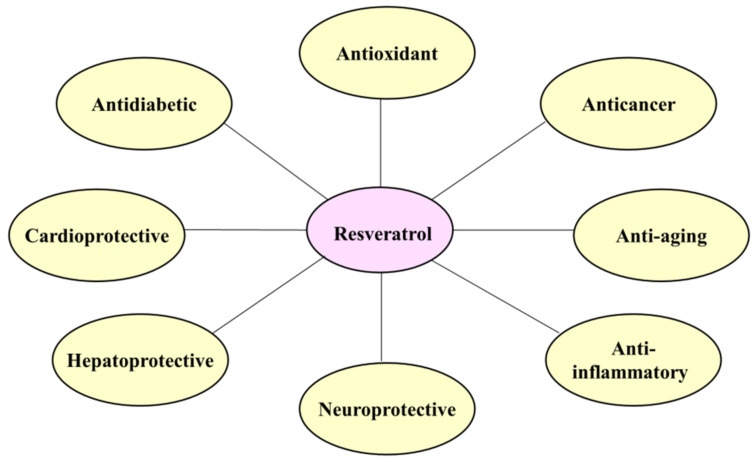
Physiological functions of resveratrol.

**Figure 3 ijms-23-13689-f003:**
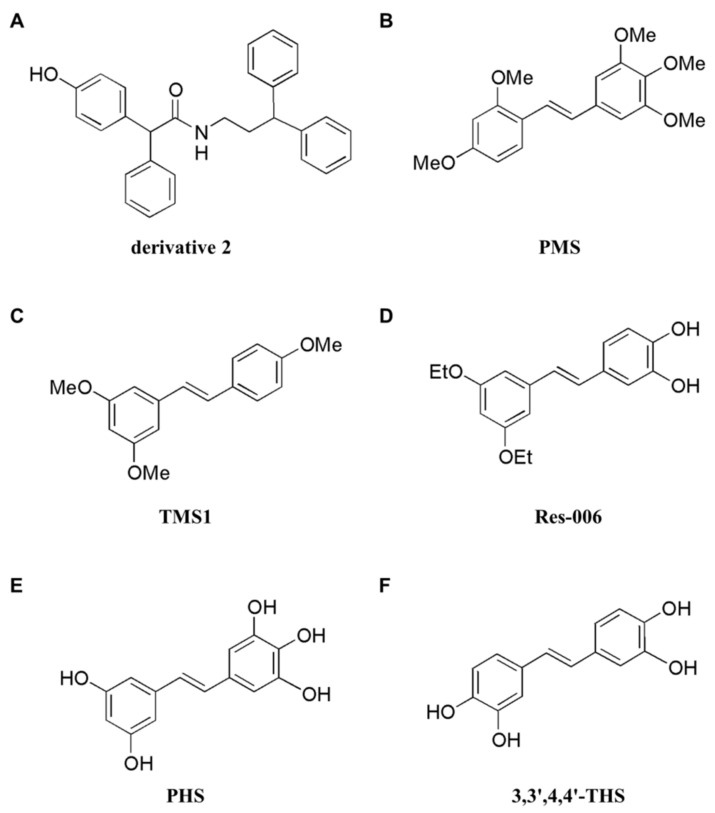
Structures of synthetic resveratrol derivatives. (**A**) derivative 2. (**B**) PMS, 2,3′,4,4′,5′-pentamethoxy-*trans*-stilbene. (**C**) TMS1, *trans*-3,5,4′-trimethoxystilbene. (**D**) Res-006, 3,5-diethoxy-3′,4′-dihydroxy-*trans*-stilbene. (**E**) PHS, 3,3′,4,5,5′-pentahydroxy-*trans*-stilbene. (**F**) 3,3′,4,4𠌩-THS, 3,3′,4,4′-tetrahydroxy-*trans*-stilbene.

**Figure 4 ijms-23-13689-f004:**
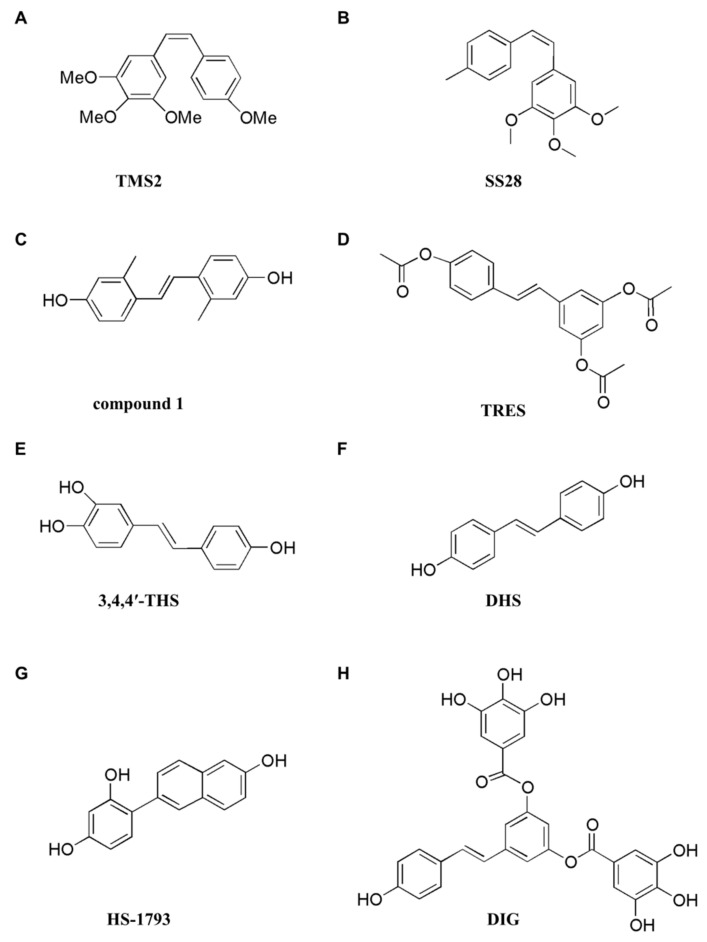
Structures of synthetic resveratrol analogues. (**A**) TMS2, (*Z*)-3,4,5,4′-*trans*-tetramethoxystilbene. (**B**) SS28, (*E*)-1,2,3-trimethoxy-5-(4-methylstyryl)benzene. (**C**) compound 1, (*E*)-4,4′-(ethene-1,2-diyl)bis(3-methylphenol). (**D**) TRES, 3,5,4′-tri-*O*-acetyl-trihydroxystilbene. (**E**) 3,4,4′-THS, 3,4,4′-trihydroxy-*trans*-stilbene. (**F**) DHS, 4,4′-dihydroxy-*trans*-stilbene. (**G**) HS-1793, 4-(6-hydroxy-2-naphthyl)-1,3-benzenediol. (**H**) DIG, 3,5-*O*-digalloyl-resveratrol. (**I**) DMU-212, 3,4,4′,5-tetramethoxystilbene. (**J**) DMU-281, 4-hydroxy-3,4,5-trimetoxystilbene. (**K**) HPIMBD, 4-(*E*)-{(4-hydroxyphenylimino)-methylbenzene, 1,2-diol}. (**L**) TIMBD, 4-(*E*)-{(p-tolylimino)-methylbenzene-1,2-diol}. (**M**) 3,4,4′-tri-MS, 3,4,4′-trimethoxy-*trans*-stilbenes. (**N**) 3,4,2′,4′-tetra-MS, 3,4,2′,4′-tetramethoxy-*trans*- stilbenes.

**Figure 5 ijms-23-13689-f005:**
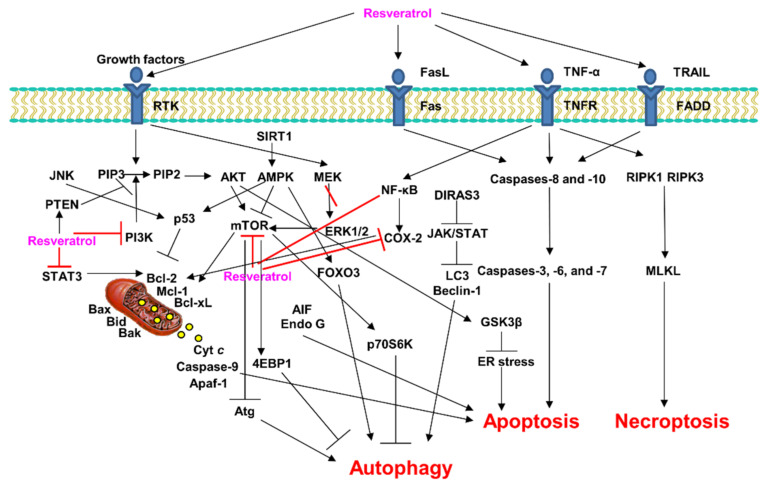
Mechanisms of PCD induction by resveratrol in cancer cells. AIF, apoptosis-inducing factor; AKT, protein kinase B; AMPK, 5′ adenosine monophosphate-activated protein kinase; Apaf-1, apoptotic protease activating factor-1; Atg, autophagy related; Bak, Bcl-2 homologous antagonist/killer; Bax, Bcl-2-associated X protein; Bcl-xL, B-cell lymphoma extra-large; Bcl-2, B-cell lymphoma-2; Bid, BH3-interacting-domain death agonist; COX-2, cyclooxygenase-2; Cyt *c*, cytochrome *c*; DIRAS3, GTP-binding protein Di-Ras3; Endo G, endonuclease G; ERK, extracellular signal-regulated protein kinases; ER stress, endoplasmic reticulum stress; FADD, FAS-associated death domain protein; FasL, apoptosis-stimulating fragment (Fas) ligand; FOXO3, forkhead box O3; GSK3β, glycogen synthase kinase 3 beta; JAK, Janus kinase; JNK, c-Jun N-terminal kinases; LC3, microtubule-associated proteins 1A/1B light chain 3; Mcl-1, myeloid cell leukemia-1; MEK, mitogen-activated protein kinase; MLKL, mixed lineage kinase domain-like protein; mTOR, mammalian target of rapamycin; NF-κB, nuclear factor kappa-light-chain-enhancer of activated B cells; PIP2, phosphatidylinositol 4,5-bisphosphate; PIP3, phosphatidylinositol (3,4,5)-trisphosphate; PI3K, phosphoinositide 3-kinase; PTEN, phosphatase and tensin homolog; p70S6K, ribosomal protein S6 kinase; RIPK, receptor-interacting serine/threonine protein kinase; RTK, receptor tyrosine kinase; SIRT, sirtuin; STAT3, signal transducer and activator of transcription 3; TNFR, tumor necrosis factor receptor; TNF-α, tumor necrosis factor-α; TRAIL, TNF-related apoptosis-inducing ligand; 4EBP1, eukaryotic translation initiation factor 4E-binding protein 1.

**Table 1 ijms-23-13689-t001:** Molecular targets of resveratrol-induced apoptosis.

Cancer/Cell Lines	Upregulation	Downregulation	Refs.
*Bladder*			
RT4		AKT, DNMT1, PLK1, mTOR, SRC	[21]
5637	caspase-3 activity	PLK1, p-AKT, Bcl-2	[21,64]
SV-HUC-1	caspase-3 activity	miR-21	[64]
T24	RASSF1A, p21, AKT, p-p38, Bax, caspase-3, -9, and PARP cleavage, caspase-3 activity	HOXB3, PLK1, cyclin D1, CDK4, p-Rb, p-AKT(Ser473), Bcl-2, Bcl-xL, p-Bad(Ser112), p-Bad(Ser136), procaspase-3 and -9, VEGF, FGF-2, miR-21	[21,64,65]
T24 xenograft		VEGF, FGF-2	[65]
*Bone*			
JJ012	SIRT1, caspase-3 cleavage	Ac-p65	[66]
JJ012 xenograft	SIRT1, caspase-3 cleavage		[66]
SW1353	Bax, caspase-3 cleavage, SIRT1	Bcl-2, p-STAT3	[67]
*Breast*			
MCF-7	caspase-8 and -9 activity, PARP cleavage	caspase-3, -8, and -9, Bcl-2, XIAP, CDK2, CDK4, CDK6, GSH, TIGAR, p-S6	[68,69]
MDA-MB-231	caspase-3, caspase-8 and -9 activity, Cyt *c*(cytosol), Bax(mitochondrial), Bim(mitochondrial), t-Bid(mitochondrial)	caspase-3, -8, and -9, Bcl-2, XIAP, CDK2, CDK4, CDK6, Bax(cytosol)	[68,70]
*Colon*			
CO115	SET7/9, Me-p53 at K372, PARP cleavage, p53		[22]
DLD1	p53, Bax	cyclin D1, cyclin E1, Bcl-2, p-STAT3(Tyr705)	[71]
HCT116	Bax, Cyt *c*, caspase-3 and -9 cleavage, SET7/9, Me-p53 at K372, PARP cleavage, p53, SIRT1	Bcl-2, procaspase-3 and -9, hERG, COX-2, EP1, EP4	[22,70,72,73,74,75]
SW480	SET7/9, Me-p53 at K372, PARP cleavage, p53, SIRT1	COX-2, EP1, EP4	[22,74,75]
SW620	Bax, Cyt *c*, caspase-3 and -9 cleavage	Bcl-2, procaspase-3 and -9	[72]
HT15	p53, Bax	cyclin D1, cyclin E1, Bcl-2, p-STAT3(Tyr705)	[71]
HT29	PARP, caspase-8, and caspase-3 cleavage, LC3-I, LC3-II	COX-2, EP1, EP4	[74,76]
COLO 201	PARP, caspase-8, and caspase-3 cleavage		[76]
*Cervical*			
HeLa	FOXO3a, Bim	p-FOXO3a, p-ERK, ∆Ψm, p53, p65	[77,78]
MES-SA		β-catenin, c-myc	[79]
CaSki		p53, p65	[78]
C33A	p53	p65	[78]
CaLo		p53, p65	[78]
Ishikawa	p-AMPKα, p-ERK, LC3, PARP cleavage		[80]
*Esophageal*			
EC109	caspase-3 cleavage, Bax, caspase-3 activity, Beclin-1, Atg5, LC3-II, p-LKB1, p-AMPK	Bcl-2, LC3-I, p-Rapter	[81]
EC9706	caspase-3 cleavage, Bax, Beclin-1, Atg5, LC3-II	Bcl-2, LC3-I	[81]
*Gastric*			
SGC7901	Bax, Bak, caspase-3, -8, and PARP1 cleavage, Cyt *c*(cytosol), MLKL, p62, VDAC1, LC3-II, Atg3, Atg5, Beclin-1, p-ERK, p-p38, BiP, CHOP, BAP31	miR-155-5p, c-Myc, cyclin B1, cyclin D1, claudin 1, Bcl-2, caspase-3 and -9, procaspase-3 and -8, p-NF-κB p65(cytoplasm), p-NF-κB p65(nucleus), p-NF-κB p65(total), ∆Ψm, Cyt *c*(mitochondria), p21, p-mTOR, p-AKT, β-catenin, Wnt3a, ZO-1, fibronectin, α-SMA, Vimentin, MMP-2	[23,82,83,84]
SGC7901 xenograft	caspase-3 cleavage	Bcl-2	[83]
MGC803		miR-155-5p	[23]
*Head and neck*			
HN3		p-STAT3(Tyr705)	[85]
FaDu	SOCS-1, caspase-3, -9, and PARP cleavage	p-STAT3(Tyr705), p-STAT3(Ser727), p-JAK2(Thy1007/1008), STAT3, Bcl-2, Bcl-xL, Survivin, IAP-1, cyclin D1, VEGF, MMP-2 and -9, procaspase-3 and -9	[85]
*Leukemia*			
U937	p-AMPK, Bax(mitochondrial)	p-AKT, Bax(cytosol), H-Ras	[86]
MOLT-4	p-AMPK, p62, LC3-II, PARP1 cleavage, caspase-3 activity, ROS	p-AKT, H-Ras, ∆Ψm	[86,87]
CCRF-CEM	Bax	Bcl-2	[88]
K562	SphK1(cytosol), ceramide	SphK1(membrane), SphK activity, S1P	[89]
HL-60	p62, LC3-I, LC3-II, caspase-3, -8, and PARP1 cleavage, caspase-3 activity, Bax, Bad, Fas, Bid, Atg5, Beclin-1, LC3II, p-AMPK, p-LKB1, p-Raptor	Bcl-2, ROS, ∆Ψm, p-Bad, FasL, PI3K(p85), p-AKT, p-p70S6K	[87,90]
*Liver*			
HepG2	Bax, PARP1 cleavage, SIRT1, SIRT1 activity, DCL1	PCNA, Bcl-2, Bcl-2/Bax ratio, PARP, caspase-3 and -7, p-PI3K, p-AKT, Ac-FOXO1, p-FOXO3a	[91]
Bel-7402	Bax, PARP1 cleavage, SIRT1, DCL1	Bcl-2, Bcl-2/Bax ratio, PARP, caspase-3 and -7, p-FOXO3a	[91]
SMMC-7721	Bax, PARP1 cleavage, SIRT1, DCL1	Bcl-2, Bcl-2/Bax ratio, PARP, caspase-3 and -7, p-FOXO3a	[91]
HL-7702	SIRT1		[91]
*Lung*			
A549	Beclin-1, LC3-I, Bax, NGFR, Ac-p53, p53, PUMA, Cyt *c*, Bak, AIF(cytosol), caspase-3 and -9 activity, Bim-L, caspase-3 cleavage, LC3-II, p62	Bcl-2, p-mTOR, p-AKT, p-NF-κB p62, Bcl-xL, ∆Ψm, AIF(mitochondrial), STAT3, p-STAT3, procaspase-3, p62, p-MDM2(Ser166), p-AKT(Ser473)	[92,93,94,95,96]
ASTC-a-1	Bak, AIF(cytosol), caspase-3 and -9 activity, Bim-L	∆Ψm, AIF(mitochondrial), Bcl-xL	[94]
H1299	PARP cleavage, LC3-II, caspase-3 cleavage	GSH, TIGAR, p-S6, caspase-3	[69]
H460	caspase-8 activity	c-FLIP, VEGF	[97]
*Melanoma*			
A375SM	p21, p27, ROS, p-eIF2α, CHOP, p-p38, p53, Bax	cyclin E, cyclin B, Nrf2, Bcl-2	[98]
*Multiple myeloma*			
U266	Beclin-1, LC3-I, LC3-II, caspase-3 and PARP cleavage, p-AMPKα	Survivin, p-mTOR, p-p70S6K, p-4EBP1	[99]
RPMI-8226	Beclin-1, LC3-I, LC3-II, caspase-3 and PARP cleavage, p-AMPKα	Survivin, p-mTOR, p-p70S6K, p-4EBP1	[99]
NCI-H929	Beclin-1, LC3-I, LC3-II, caspase-3 and PARP cleavage, p-AMPKα	Survivin, p-mTOR, p-p70S6K, p-4EBP1	[99]
*Oral*			
CAR	AMPKα, p-AMPKα(Thr172), Atg5, Atg7, Atg12, Atg14, Atg16L1, Beclin-1, PI3K class III, LC3-II, caspase-3 and -9 cleavage, Cyt *c*, Apaf-1, AIF, Endo G, Bax, Bad, caspase-3 and -9 activity	p-AKT(Ser473), p-mTOR(Ser2448), Rubicon, Bcl-2, p-Bad(Ser136)	[100]
CAL27	Bak, Bax, Apaf-1, caspase-3, ICAD and PARP cleavage, E-cadherin, N-cadherin	Bcl-2, Bcl-xL, procaspase-3 and -9, Snail, Slug, Smad2/3	[101]
HSC-3	PARP and caspase-3 cleavage, caspase-3 and -9 activity, p16	CBX7, p-AKT	[102]
*Ovarian*			
OVCAR-3	Atg5, p62, LC3-II, caspase-3 and PARP activity, ROS	∆Ψm	[103]
Caov-3	LC3-II, p62		[103]
A2780	p-AMPK, caspase-3 cleavage	p-mTOR	[104]
SKOV3	p-AMPK, caspase-3 cleavage, miR-34a, Bax	p-mTOR, Bcl-2	[104,105]
SKOV3 xenograft		Ki-67 index, Metastasis index	[104]
OV-90	miR-34a, Bax, caspase-3 cleavage	Bcl-2	[105]
*Pancreatic*			
MIA PaCa-2		IHH, Ptch, SMO	[106]
*Prostate*			
C4-2B	PARP, Bax, Bid, Bak, p27, p53	Bcl-2, Mcl-1, Bcl-xL, CDK1, CDK2, CDK4, cyclin D1, cyclin E1	[107]
DU145	PARP, Bax, Bid, p27, p53	Bcl-2, Mcl-1, Bcl-xL, CDK1, CDK2, CDK4, cyclin E1, PCNA, cyclin B1	[107]
LNCaP	caspase-3 activity	p-PI3K, p-AKT, p-mTOR, p-FKHR, p-FKHRL1, p-AFX	[70,108]
PC-3	Bax, caspase-3 and -9 cleavage, E-cadherin, caspase-3 activity	Bcl-2, ∆Ψm, Vimentin, p-AKT, pAKT/AKT	[70,109]
*Thyroid*			
BHP 18–21	p-ERK1, p-ERK2, p53, p-p53(ser15), p21, c-Fos		[110]
BHP 2–7	p-ERK1, p-ERK2, p53, p21, c-Fos, c-Jun		[110]
FTC 236	p-ERK1, p-ERK2, p53, p-p53(ser15), c-Fos, p21		[110]
FTC 238	c-Fos, p53, p21		[110]
THJ-16T	caspase-3 and -9 activity	SOD2, CAT, procaspase-3 and -9	[111]
THJ-11T	SULT1A1		[111]

Ac-, acetylation; AIF, apoptosis-inducing factor; AFX (FOXO4), forkhead box protein O4; AKT, protein kinase B; Apaf-1, apoptotic protease activating factor-1; AMPK, 5′ adenosine monophosphate-activated protein kinase; Atg, autophagy related; Bad, Bcl-2-associated death promoter; Bak, Bcl-2 homologous antagonist/killer; BAP31, B-cell receptor-associated protein 31; Bax, Bcl-2-associated X protein; Bcl-2, B-cell lymphoma-2; Bcl-xL, B-cell lymphoma extra-large; Bid, BH3-interacting-domain death agonist; Bim, Bcl-2-interacting mediator of cell death; Bim-L, Bim-long; BiP, binding immunoglobulin protein; CAT, catalase; CBX7, chromobox protein homolog 7; CDK, cyclin-dependent kinase; c-FLIP, cellular FLICE (FADD-like IL-1β-converting enzyme)-inhibitory protein; CHOP, CCAAT/enhancer-binding protein (C/EBP) homologous protein; COX-2, cyclooxygenase-2; Cyt *c*, cytochrome *c*; DLC1, deleted in liver cancer 1; DNMT1, DNA (cytosine-5)-methyltransferase 1; DR, death receptor; E-cadherin, epithelial cadherin; N-cadherin, neural cadherin; eIF2α, eukaryotic initiation factor 2 alpha; Endo G, endonuclease G; EP, prostaglandin E2 receptor; ERK, extracellular signal-regulated protein kinases; FasL, apoptosis-stimulating fragment (Fas) ligand; FGF-2, fibroblast growth factor-2; FKHR (FOXO1), forkhead in rhabdomyosarcoma; FKHRL1 (FOXO3), forkhead transcription factor-like 1; FOXO, forkhead box O; GSH, glutathione; hERG, human ether-a-go-go-related gene; HOXB3, homeobox protein Hox-B3; IAP, inhibitor of apoptosis; ICAD, inhibitor of caspase-activated DNase; IHH, Indian hedgehog; JAK2, Janus kinase 2; LC3, microtubule-associated proteins 1A/1B light chain 3; LKB1, liver kinase B1; Mcl-1, myeloid cell leukemia-1; MDM2, mouse double minute 2; miR, microRNA; Me-, methylation; MLKL, mixed lineage kinase domain-like protein; mTOR, mammalian target of rapamycin; MMP-2, matrix metalloproteinase-2; NF-κB p65, nuclear factor kappa-light-chain-enhancer of activated B cells p65; NGFR, nerve growth factor receptor; Nrf2, nuclear factor erythroid 2-related factor 2; p-, phosphorylation; PARP, poly(ADP-ribose) polymerase; PCNA, proliferating cell nuclear antigen; PLK-1, polo-like kinase 1; Ptch, patched homolog; PI3K, phosphoinositide 3-kinase; PUMA, p53-upregulated modulator of apoptosis; p70S6K, ribosomal protein S6 kinase; RASSF1A, Ras association domain family 1 isoform A; Rb, retinoblastoma protein; ROS, reactive oxygen species; Rubicon, run domain Beclin-1-interacting and cysteine-rich domain-containing protein; SET7/9, SET domain containing lysine methyltransferase 7/9; SIRT, sirtuin; S6, ribosomal protein; SOCS-1, suppressor of cytokine signaling 1; SOD2, superoxide dismutase 2; SphK1, sphingosine kinase 1; Src, proto-oncogene tyrosine-protein kinase; SMO, smoothened homolog; STAT3, signal transducer and activator of transcription 3; S1P, sphingosine-1-phosphate; SULT1A1, sulfotransferase isoform 1A1; t-, truncated; TIGAR, *TP53*-induced glycolysis and apoptosis regulator; TUNEL, terminal deoxynucleotidyl transferase dUTP nick end labeling; VDAC1, voltage-dependent anion channel 1; VEGF, vascular endothelial growth factor; XIAP, X-linked inhibitor of apoptosis protein; 4E-BP1, eukaryotic translation initiation factor 4E-binding protein 1; ZO-1, zonula occludens-1; α-SMA, α-smooth muscle actin; ∆Ψm, mitochondrial membrane potential.

**Table 2 ijms-23-13689-t002:** Molecular targets of resveratrol-induced autophagy.

Cancer/Cell Lines	Upregulation	Downregulation	Refs.
*Breast*			
MCF-7	PARP cleavage, Beclin-1, Atg7, LC3-II	GHS, TIGAR, p-S6, LC3-I, β-catenin, cyclin D1	[69,128]
SUM-159	Beclin-1, Atg7, LC3-II	LC3-I, cyclin D1	[128]
*Colon*			
HT29	caspase-3, -8, and PARP cleavage, LC3-I, LC3-II		[76]
COLO 201	caspase-3, -8, and PARP cleavage		[76]
*Cervical*			
Ishikawa	p-AMPKα, p-ERK, LC3, PARP cleavage		[80]
*Esophageal*			
EC109	caspase-3 cleavage, Bax, caspase-3 activity, Beclin-1, Atg5, LC3-II, p-LKB1, p-AMPK	Bcl-2, LC3-I, p-Rapter	[81]
EC9706	caspase-3 cleavage, Bax, Beclin-1, Atg5, LC3-II	Bcl-2, LC3-I	[81]
*Gastric*			
SGC7901	Bak, Bax, MLKL, p62, VDAC1, LC3-II, Atg3, Beclin-1, p-ERK, p-p38, BiP, CHOP, BAP31	cyclin B1, p21, p-mTOR, p-AKT, β-catenin, Wnt3a, ZO-1, fibronectin, α-SMA, Vimentin, MMP-2	[84]
*Leukemia*			
MOLT-4	p62, LC3-II, PARP1 cleavage, caspase-3 activity, ROS	∆Ψm	[87]
HL-60	p62, LC3-I, LC3-II, caspase-3, -8, and PARP1 cleavage, caspase-3 activity, Bax, Bad, Fas, Bid, Atg5, Beclin-1, LC3II, p-AMPK, p-LKB1, p-Raptor	Bcl-2, ROS, ∆Ψm, p-Bad, FasL, PI3K(p85), p-AKT, p-p70S6K	[87,90]
K562	p62, LC3-I, LC3-II, p-JNK2/3(Thr 183/Tyr 185), p-c-Jun(Ser 63), p-AMPKα(Thr 172)	p-mTOR(Ser )2448, p-p70/85-S6K(Thr389 ), p-S6 ribo(Ser 235/236), p-4EBP1(Thr 37/46)	[129]
*Liver*			
MHCC-97H	Beclin-1, LC3-II, LC3-II/I, p53	p62, p-AKT, p-AKT/AKT	[130]
*Lung*			
A549	Beclin-1, LC3-I, LC3-II, Bax, NGFR, caspase-3 and -8 cleavage, p53, Ac-p53, p53, PUMA, Cyt *c*, Cyt *c*(cytosol), Atg5, p-Raptor, p-AMPK, p62, SIRT1, LC3-II/LC3-I, p-p38, p-p38/p38, caspase-3 activity, Fas, Cav-1	Bcl-2, p-mTOR, procaspase-3, p62, p-MDM2(Ser166), p-AKT(Ser473), p62, Bcl-xL, LC3-I, p-p70S6K, p-AKT/AKT, p-p70S6K/p70S6K, p-mTOR/mTOR, Cyt *c*(mitochondrial),	[92,93,96,131,132,133]
H1299	PARP cleavage, LC3-II, caspase-3 cleavage	GSH, TIGAR, p-S6, caspase-3	[69]
*Multiple myeloma*			
U266	Beclin-1, LC3-I, LC3-II, caspase-3 and PARP cleavage, p-AMPKα	Survivin, p-mTOR, p-p70S6K, p-4EBP1	[99]
RPMI-8226	Beclin-1, LC3-I, LC3-II, caspase-3 and PARP cleavage, p-AMPKα	Survivin, p-mTOR, p-p70S6K, p-4EBP1	[99]
NCI-H929	Beclin-1, LC3-I, LC3-II, caspase-3 and PARP cleavage, p-AMPKα	Survivin, p-mTOR, p-p70S6K, p-4EBP1	[99]
*Oral*			
CAR	AMPKα, p-AMPKα(Thr172), Atg5, Atg7, Atg12, Atg14, Atg16L1, Beclin-1, PI3K class III, LC3-II, caspase-3 and -9 cleavage, Cyt *c*, Apaf-1, AIF, Endo G, Bax, Bad, caspase-3 and -9 activity	p-AKT(Ser473), p-mTOR(Ser2448), Rubicon, Bcl-2, p-Bad(Ser136)	[100]
*Ovarian*			
OVCAR-3	Atg5, p62, LC3-II, caspase-3 and PARP activity, ROS, ARHI, Beclin-1	∆Ψm, p-AKT(Ser473), p-S6 (Ser235/236), p-STAT3(Thy705)	[103,134,135]
Caov-3	LC3-II, p62	p-STAT3	[103,135]
SKOV-3	Beclin-1, LC3-II		[136]

Ac-, acetylation; AIF, apoptosis-inducing factor; AKT, protein kinase B; Apaf-1, apoptotic protease activating factor-1; AMPK, 5′ adenosine monophosphate-activated protein kinase; ARHI, aplasia ras homology member I; Atg, autophagy related; Bad, Bcl-2-associated death promoter; Bak, Bcl-2 homologous antagonist/killer; BAP31, B-cell receptor-associated protein 31; Bax, Bcl-2-associated X protein; Bcl-2, B-cell lymphoma-2; Bcl-xL, B-cell lymphoma extra-large; Bid, BH3-interacting-domain death agonist; BiP, binding immunoglobulin protein; Cav-1, caveolin-1; CHOP, CCAAT/enhancer-binding protein (C/EBP) homologous protein; Cyt *c*, cytochrome *c*; Endo G, endonuclease G; ERK, extracellular signal-regulated protein kinases; Fas, apoptosis-stimulating fragment; FasL, apoptosis-stimulating fragment (Fas) ligand; GSH, glutathione; JNK, c-Jun N-terminal kinases; LC3, microtubule-associated proteins 1A/1B light chain 3; LKB1, liver kinase B1; MDM2, mouse double minute 2; MLKL, mixed lineage kinase domain-like protein; MMP-2, matrix metalloproteinase-2; mTOR, mammalian target of rapamycin; NGFR, nerve growth factor receptor; p-, phosphorylation; PARP, poly(ADP-ribose) polymerase; PI3K, phosphoinositide 3-kinases; PUMA, p53-upregulated modulator of apoptosis; p70S6K, ribosomal protein S6 kinase; ROS, reactive oxygen species; SIRT, sirtuin; STAT3, signal transducer and activator of transcription 3; S6, ribosomal protein; TIGAR, *TP53*-induced glycolysis and apoptosis regulator; VDAC1, voltage-dependent anion channel 1; ZO-1, zonula occludens-1; 4EBP1, eukaryotic translation initiation factor 4E-binding protein 1; α-SMA, α-smooth muscle actin; ∆Ψm, mitochondrial membrane potential.

**Table 3 ijms-23-13689-t003:** Molecular targets of synthetic resveratrol derivatives and analogues for the induction of programmed cell death.

Type	Name	Mechanism	Upregulation	Downregulation	Cancer/Cell Lines	Refs.
*Derivatives*	derivative 2	apoptosis	Bax, PARP cleavage	cyclin D1, CDK4, Bcl-2	Breast (MCF-7, MDA-MB-231)	[160]
	PMS	apoptosis	caspase-3, -7, -9, and PARP cleavage, Bad, Bik, Bok, Bim, PUMA, Bcl-2, Bcl-2(Thr56), Bcl-2(Ser70)	Mcl-1	Colon (HT29)	[161]
	TMS1	apoptosis	p53, p-H2AX(Ser139), p-p53(Ser15), p-p53(Ser46), p- p53(Ser392), PUMA		Osteosarcoma (143B cell)	[156]
	Res-006	apoptosis	caspase-3 and PARP cleavage, p-IRE1α, p-JNK, XBP1, ERdj4, p-PERK, p-elF2α, CHOP, 3XFlag-ATF6α∆C, GRP78		Liver (HepG2)	[162]
	PHS	apoptosis	caspase-3, -9, and PARP cleavage, Bad	p-AKT(Ser473), GSH	Colon (HT29)	[163]
	3,3′,4,4′ -THS	apoptosis	ROS, 8-OH-dG, hOGG1, caspase-3, -8, and -9 activity, SA-*β*-gal activity, p-p38/p38	SOD, CAT	Ovarian (A2780, OVCAR-3, SKOV-3)	[164]
*Analogues*	TMS2	apoptosis, autophagy	caspase-3 and PARP cleavage, p-PERK, CHOP, p-eIF2α, p-AMPK, LC3-II, p-JNK	Bcl-2, p-AKT, p-p70S6K, p-S6, p-ACC, p-EGFR, p-PI3K, p-ERK	Lung (H1975)	[165]
	SS28	apoptosis	caspase-3, -9, and PARP cleavage	∆Ψm, cyclin B1, CDK6	Lung (A549), Leukemia (CEM)	[166]
	compound 1	apoptosis	ROS, p53, p21, Bax	cyclin A1, cyclin A2, Bcl-2	Cervical (HeLa)	[167]
	TRES	apoptosis	caspase-3 and PARP cleavage, Bim, PUMA, p-STAT3(cytoplasm), p-NF-κB(cytoplasm)	Mcl-1, p-STAT3, p-NF-κB, p-STAT3(nucleus), p-NF-κB(nucleus)	Pancreatic (PANC-1, BxPC-3)	[168]
	3,4,4′-THS	apoptosis, autophagy	caspase-3, -9, and PARP cleavage, Bax, LC3-II, ROS	Bcl-2, Survivin, p62, p-p70S6K, p-4EBP1	Lung (A549)	[169]
	DHS	apoptosis	PARP-1 cleavage		Lung (LLC)	[170]
	HS-1793	apoptosis	p53, p21, Fas-L, Fas, PARP cleavage, Bax, caspase-3, -8, and -9 activity, ERK, p-ERK, JNK, p-JNK	MDM2, cyclin D1, CDK4, cyclin B1, Cdc2, Cdc25C, Bcl-2	Breast (MCF-7, MDA-MB-231)	[171]
		apoptosis	caspase-3, -8, and PARP cleavage, Bax, Cyt *c*(cytosol)	procaspase-3, -8, and -9, Bcl-2, Cyt *c*(mitochondria), cyclin B1, Cdc2, Cdc25C, CDK2, CDK4, CDK6, p-AKT, p-p38, p-ERK1/2, p-JNK	Colon (HCT116)	[172]
		apoptosis	PARP cleavage, CHOP, GRP78	procaspase-3, XBP1, p-AKT	Colon (HT29)	[173]
		apoptosis	PARP cleavage	caspase-3, -6, Mcl-1, Bcl- 2, Bcl-xL, XIAP, 14-3-3,p-Bad, (Ser136), p-Bad(Ser155)	Leukemia (U937)	[174]
	DIG	apoptosis	p-Chk2(Thr68), p-Cdc25A(Ser177), p-ATM(Ser1981), p-p38(Thr180/Tyr182), p-AKT (Ser473)		Pancreatic (AsPC-1)	[175]
	DMU-212	apoptosis	EF value, caspase-3/7 and -9 activity, Bax, Apaf-1, p53, Bad, Bak1, Bik, Bok, Noxa, PARP-1 cleavage	Bag1, Bcl-2, Bcl-xL, CYP1A1, CYP1B1	Colon (DLD-1, LOVO)	[176]
		apoptosis	EF value, caspase-3/7 activity, Fas, FasL, TNF, TNFRF10A, TNFRSF21, TNFRSF16, Bax, Apaf-1, p53	TRAF-1, -3, -5, and -7, BIRC-2, Bcl-2, Bcl2l10, CYP1A1, CYP1B1	Ovarian (A-2780, SKOV-3)	[177]
	DMU-281	apoptosis	caspase-3/7, -8, and -9 activity, Bik, Bad, Bak1, Fas, TNFSRF10B, TNFSRF11B, TNFSF8, FADD, HSP60	Bcl-2, Bcl2L1, HMGB1, STAT5A, STAT5b, HSP27	Colon (LoVo)	[178]
		apoptosis	Bik, TNF, caspase-3/7, -8, and -9 activity, Smac/Diablo	Bcl-2, Bcl-xL, BIRC2, HMGB1, STAT5b, TNFRSF10C, TNFRSF11B, TRAF-1, -3, and -5 procaspase-3, HSP27	Colon (DLD-1)	[178]
	HPIMBD	apoptosis, autophagy	Beclin-1, LC3-I, LC3-II	ERα, c-Myc	Breast (MDA-MB-231, T47D)	[179]
	TIMBD	apoptosis, autophagy	Beclin-1, LC3-I, LC3-II	ERα, c-Myc	Breast (MDA-MB-231, T47D)	[179]
	3,4,4ʹ-tri-MS, 3,4,2ʹ,4ʹ-tetra-MS	apoptosis	p53, Bax	Bcl-xL	Leukemia (HL-60, THP-1)	[180]

ACC, acetyl-CoA carboxylase; AKT, protein kinase B; AMPK, 5′ adenosine monophosphate-activated protein kinase; Apaf-1, apoptotic protease activating factor-1; ATM, ataxia telangiectasia mutated kinase; Bad, Bcl-2-associated death promoter; Bag1, Bcl-2-associated athanogene-1; Bak, Bcl-2 homologous antagonist/killer; Bax, Bcl-2-associated X protein; Bcl-2, B-cell lymphoma-2; Bcl2L1, Bcl-2-like protein 1; Bcl2l10, Bcl-2-like protein 10; Bcl-xL, B-cell lymphoma extra-large; Bik, Bcl-2-interacting killer; Bim, Bcl-2-interacting mediator of cell death; BIRC2, baculoviral IAP repeat-containing 2; Bok, Bcl-2 related ovarian killer; CAT, catalase; Cdc2, cell division control protein 2; Cdc25, cell division cycle 25; CDK, cyclin-dependent kinase; CHOP, CCAAT/enhancer-binding protein (C/EBP) homologous protein; compound 1, (*E*)-4,4′-(ethene-1,2-diyl)bis(3-methylphenol); Chk2, checkpoint kinase 2; CYP1A1, cytochrome P450, family 1, subfamily A, polypeptide 1; DHS, 4,4′-dihydroxy-*trans*-stilbene; DIG, 3,5-*O*-digalloyl-resveratrol; DMU-212, 3,4,4′,5-tetramethoxystilbene; DMU-281, 4′-hydroxy-3,4,5-trimetoxystilbene; EF, enrichment factor; EGFR, epidermal growth factor receptor; eIF2α, eukaryotic initiation factor 2 alpha; ERdj4, endoplasmic reticulum–localized DnaJ 4; ERK, extracellular signal-regulated protein kinases; ERα, estrogen receptor alpha; FADD, FAS-associated death domain protein; FasL, apoptosis-stimulating fragment (Fas) ligand; GRP78, glucose-related protein 78; GSH, glutathione; HMGB1, High mobility group box 1; hOGG1, human 8-oxoguanine DNA N-glycosylase 1; HSP, heat shock protein; HPIMBD, 4-(*E*)-{(4-hydroxyphenylimino)-methylbenzene, 1,2-diol}; HS-1793, 4-(6-Hydroxy-2-naphthyl)-1,3-benzenediol; H2AX, H2A histone family member X; IRE1α, inositol-requiring 1α; JNK, c-Jun N-terminal kinases; LC3, microtubule-associated proteins 1A/1B light chain 3; Mcl-1, myeloid cell leukemia-1; MDM2, mouse double minute 2; NF-κB, nuclear factor kappa-light-chain-enhancer of activated B cells; Noxa, phorbol-12-myristate-13-acetate-induced protein 1; p-, phosphorylation; PARP, poly(ADP-ribose) polymerase; PERK, protein kinase-like endoplasmic reticulum kinase; PHS, 3,3′,4,5,5′-pentahydroxy-*trans*-stilbene; PI3K, phosphoinositide 3-kinases; PMS, 2,3′,4,4′,5′-pentamethoxy-*trans*-stilbene; PUMA, p53-upregulated modulator of apoptosis; p70S6K, ribosomal protein S6 kinase; Res-006, 3,5-diethoxy-3′,4′-dihydroxy-*trans*-stilbene; ROS, reactive oxygen species; SA-β-gal, senescence-associated beta-galactosidase; Smac/Diablo, second mitochondria-derived activator of caspases/Diablo homolog; SOD, superoxide dismutase; STAT, signal transducer and activator of transcription; S6, ribosomal protein; SS28, (*E*)-1,2,3-trimethoxy-5-(4-methylstyryl)benzene; TIMBD, 4-(*E*)-{(p-tolylimino)-methylbenzene-1,2-diol}; TMS1, *trans*-3, 5, 4′-trimethoxystilbene; TMS2, (*Z*)-3,4,5,4′-*trans*-tetramethoxystilbene; TNF, tumor necrosis factor; TNFSF8, tumor necrosis factor ligand superfamily member 8; TNFRSF, tumor necrosis factor receptor superfamily member; TRAF, TNF-receptor-associated factor; TRES, 3,5,4′-tri-*O*-acetyl-trihydroxystilbene; XBP1, X-box binding protein 1; XIAP, X-linked inhibitor of apoptosis protein; 3,3′,4,4′-THS, 3,3′,4,4′-tetrahydroxy-*trans*-stilbene; 3,4,2′,4′-tetra-MS, 3,4,2′,4′-tetramethoxy-*trans*-stilbene; 3,4,4′-tri-MS, 3,4,4′-trimethoxy-*trans*-stilbene; 3,4,4′-THS, 3,4,4′-trihydroxy-*trans*-stilbene; 3XFlag-ATF6α∆C, S1P and S2P protease-mediated cleavage fragment; 4EBP1, eukaryotic translation initiation factor 4E-binding protein 1; 8-OH-dG, 8-hydroxy-2′-deoxyguanosine; ∆Ψm, mitochondrial membrane potential.

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
