# Peer review of "Mechanism of Resveratrol-Induced Programmed Cell Death and New Drug Discovery against Cancer: A Review"

_ijms, 2022, doi:10.3390/ijms232213689_

Round 1

Reviewer 1 Report

Dear Authors,

tha manuscript "Mechanism of Resveratrol-Induced Programmed Cell Death and New Drug Discovery against Cancer: A Review" describes the effect of resveratrol, its derivatives and analogues in different types of cancer cells with their mechanism of action. The work has merit and can be a valuable source of information about anticancer action of these compounds. I have a few minor comments:

1. In the Abstract - line 23 - should it be "anti-human cancer drugs"? Please correct.

2. In line - 690 should it be "DMU-218" or "DMU-281"?

3. It would be also good to present the main ways of action of resveratrol in cancer cells on some diagram.

Author Response

Response to Reviewer 1 Comments

Dear editors and reviewers:

Thank you for your letter and for the reviewers’ comments concerning our manuscript entitled “Mechanism of Resveratrol-Induced Programmed Cell Death and New Drug

Discovery against Cancer: A Review” (ID: ijms-1992930). These comments are all valuable and very helpful for revising and improving our paper, as well as the important guiding significance to our researches. We have studied comments carefully and have made corrections which we hope meet with approval. Revised portions are marked in red on the paper. The main corrections in the paper and the response to the reviewer’s comments are as flowing:

  • Point 1: In the Abstract - line 23 - should it be "anti-human cancer drugs"? Please correct.

Response 1: Thanks for your comment. We edited the manuscript.

  • Point 2: In line - 690 should it be "DMU-218" or "DMU-281"?

Response 2: We are very sorry for the mistake. We edited the manuscript.

  • Point 3: It would be also good to present the main ways of action of resveratrol in cancer cells on some diagram.

Response 3: Thanks for your suggestion. We added to the manuscript the mechanism of resveratrol-induced PCD induction in cancer cells (Figure 5).

We tried our best to improve the manuscript and made some changes in the manuscript. These changes will not influence the content and framework of the paper. And here we did not list the changes but marked them in red in the revised paper. We appreciate for Editors/Reviewers’ warm work earnestly and hope that the correction will meet with approval.

Once again, thank you very much for your comments and suggestions.

Reviewer 2 Report

The manuscript ‘Mechanism of Resveratrol-Induced Programmed Cell Death and New Drug Discovery against Cancer: A Review’ needs a major revision.

 Please add objectives as a paragraph at the end of the introduction.

Please summarize in the introduction what is new in the current review compared to other recent review papers, for example, the one published in resveratrol and cancer ‘Potential therapeutic targets of resveratrol, a plant polyphenol, and its role in the therapy of various types of cancer’.

Please summarize in the introduction part, the main recent progress in the therapeutic of resveratrol.

 The Figures quality of structures synthetic resveratrol is low, and it is better to re-draw with a specific software

In the conclusions add and future perspective.

Author Response

Response to Reviewer 2 Comments

Dear editors and reviewers:

Thank you for your letter and for the reviewers’ comments concerning our manuscript entitled “Mechanism of Resveratrol-Induced Programmed Cell Death and New Drug

Discovery against Cancer: A Review” (ID: ijms-1992930). These comments are all valuable and very helpful for revising and improving our paper, as well as the important guiding significance to our researches. We have studied comments carefully and have made corrections which we hope meet with approval. Revised portions are marked in red on the paper. The main corrections in the paper and the response to the reviewer’s comments are as flowing:

  • Point 1: Please add objectives as a paragraph at the end of the introduction.

Response 1: Thanks for your comment. At the end of the introduction to our manuscript, the objectives were written as you suggested (Page 2, Lines 51-53).

  • Point 2: Please summarize in the introduction what is new in the current review compared to other recent review papers, for example, the one published in resveratrol and cancer ‘Potential therapeutic targets of resveratrol, a plant polyphenol, and its role in the therapy of various types of cancer’.

Response 2: Thanks for the advice. Unlike previous review papers, our review also included information on the programmed cell death in cancer cells by resveratrol as well as its synthetic derivatives and analogues. These contents are described at the end of the introduction (Page 2, Lines 51-53).

  • Point 3: Please summarize in the introduction part, the main recent progress in the therapeutic of resveratrol.

Response 3: Thanks for your suggestion. The introduction of the manuscript was revised by adding recent clinical studies using resveratrol (Page 3, Lines 110-122).

  • Point 4: The Figures quality of structures synthetic resveratrol is low, and it is better to re-draw with a specific software

Response 4: Thanks for the advice. We have improved the quality of the figures in the manuscript.

  • Point 5: In the conclusions add and future perspective.

Response 5: Thank you for your comments. We have revised the conclusion of the manuscript (Pages 24-25, Lines 737-742).

We tried our best to improve the manuscript and made some changes in the manuscript. These changes will not influence the content and framework of the paper. And here we did not list the changes but marked them in red in the revised paper. We appreciate for Editors/Reviewers’ warm work earnestly and hope that the correction will meet with approval.

Once again, thank you very much for your comments and suggestions.

Round 2

Reviewer 2 Report

The manuscript could be accepted in this present form.